# Microglia *TREM2^R47H* Alzheimer-linked variant enhances excitatory transmission and reduces LTP via increased TNF-α levels

Siqiang Ren[1,2†], Wen Yao[1,2†], Marc D Tambini[1,2], Tao Yin[1,2], Kelly A Norris[1,2], Luciano D'Adamio[1,2*]

[1]Department of Pharmacology, Physiology and Neuroscience, New Jersey Medical School, Newark, United States; [2]Brain Health Institute, Jacqueline Krieger Klein Center in Alzheimer's Disease and Neurodegeneration Research, Rutgers, The State University of New Jersey, Newark, United States

**Abstract** To study the mechanisms by which the p.R47H variant of the microglia gene and Alzheimer's disease (AD) risk factor TREM2 increases dementia risk, we created *Trem2^R47H* KI rats. *Trem2^R47H* rats were engineered to produce human Aβ to define human-Aβ-dependent and -independent pathogenic mechanisms triggered by this variant. Interestingly, pre- and peri-adolescent *Trem2^R47H* rats present increased brain concentrations of TNF-α, augmented glutamatergic transmission, suppression of Long-term-Potentiation (LTP), an electrophysiological surrogate of learning and memory, but normal Aβ levels. Acute reduction of TNF-α activity with a neutralizing anti-TNF-α antibody occludes the boost in amplitude of glutamatergic transmission and LTP suppression observed in young *Trem2^R47H/R47H* rats. Thus, the microglia-specific pathogenic *Trem2* variant boosts glutamatergic neuronal transmission and suppresses LTP by increasing brain TNF-α concentrations, directly linking microglia to neuronal dysfunction. Future studies will determine whether this phenomenon represents an early, Aβ-independent pathway that facilitates dementia pathogenesis in humans.

*For correspondence:
luciano.dadamio@rutgers.edu

†These authors contributed equally to this work

**Competing interests:** The authors declare that no competing interests exist.

## Introduction

AD, the most common progressive neurodegenerative disorder in the elderly, is diagnosed when clinical dementia occurs concomitantly with Aβ plaques, neurofibrillary tangles and neuronal loss (*James et al., 2014*). Genome-wide association studies have discovered *Triggering Receptor Expressed on Myeloid Cells 2* (*TREM2*) human variants that increase the risk of developing sporadic AD (*Guerreiro et al., 2013*). In the central nervous system (CNS), *TREM2* is exclusively expressed in microglia (*Schmid et al., 2002*). This genetic evidence directly implicates microglia function in AD pathogenesis.

Microglia surround amyloid plaques both in AD patients (*McGeer et al., 1987*) and Aβ plaques-bearing mice (*Frautschy et al., 1998*). Recent data suggest that these disease-associated microglia (DAM) possess enhanced activities, including the Aβ plaque-clearing activity (*Keren-Shaul et al., 2017*; *Mazaheri et al., 2017*). Through diverse mechanisms (*Kleinberger et al., 2017*; *Mazaheri et al., 2017*; *Schlepckow et al., 2017*; *Song et al., 2018*; *Ulland et al., 2015*), disease-associated TREM2 variants cause a loss of function of TREM2 that inhibits microglia transition to DAMs and impairs Aβ plaque-clearing activities (*Mazaheri et al., 2017*). As for the AD-associated p.R47H *TREM2* variant, in vitro studies suggest that it destroys an essential lipid and Aβ-binding site

within the TREM2 ectodomain, reducing the Aβ-phagocytosis capabilities of microglia (*Yeh et al., 2016*).

Model organisms are useful tools to study how human pathogenic mutation/variants alter protein's functions and promote disease in humans; thus, to dissect the pathogenic mechanisms of the p.R47H *TREM2* variant, we generated *Trem2*<sup>R47H</sup> knock-in (KI) rats, which carry the p.R47H variant in the rat endogenous *Trem2* gene (*Tambini and D'Adamio, 2020*). Rat and human APP differ by 3 amino acids in the Aβ region. These differences may be crucial since human Aβ may have higher propensity to form toxic species as compared to rodent Aβ and the pathogenic role of the p.R47H *TREM2* variant may be linked to toxic Aβ clearance deficits. To eliminate this potential issue, together with the *Trem2* mutation we introduced mutations to 'humanize' the rat Aβ sequence (*App*<sup>h</sup> allele) (*Tambini et al., 2019*). In *Trem2*<sup>R47H</sup> KI rats, transcription, splicing and translation of this pathogenic variant is controlled by endogenous regulatory elements, allowing to study pathogenic mechanisms triggered by the p.R47H *TREM2* variant in a model organism mimicking the genetics of the human disease and expressing physiological levels of human Aβ.

Pre-adolescent (*Sengupta, 2013*) *Trem2*<sup>R47H</sup> rats showed no significant alterations in brain levels of human Aβ40 and Aβ42, the latter is considered the pathogenic Aβ species (*Tambini and D'Adamio, 2020*). In addition, the Aβ42/Aβ40 ratio, another indicator of Aβ-mediated pathogenesis, is not altered. Thus, it is likely that the consequences of the Aβ-clearance deficits caused by the p.R47H *TREM2* variant fully manifest in vivo in an aging-dependent manner. Yet, the p.R47H *TREM2* may trigger both human Aβ-dependent and Aβ-independent pathogenic mechanism. Aβ-independent mechanism may precede and, perhaps, participate in mechanisms leading to dementia.

In macrophages, TREM2 functions to inhibit pro-inflammatory cytokines production, especially TNF-α (*Turnbull et al., 2006*). Thus, it is possible that loss of TREM2 function caused by the p.R47H variant may favor pro-inflammatory cytokine production by microglia. In this study, we tested this hypothesis in young *Trem2*<sup>R47H</sup> rats with the purpose of determining potential early pathogenic mechanisms caused by the p.R47H variant.

## Results

### Increased concentration of TNF-α and other pro-inflammatory cytokines in the CNS and CSF of young animals carrying the *Trem2*<sup>R47H</sup> variant

Pre-adolescent (4 weeks old) *Trem2*<sup>R47H</sup> rats showed no significant alterations in CNS levels of human Aβ40, Aβ42 and the Aβ42/Aβ40 ratio (*Tambini and D'Adamio, 2020*), even though the Trem2<sup>R47H</sup> variant reduces binding and clearance of human Aβ in vitro (*Zhao et al., 2018*). This discrepancy prompted us to assess further Aβ metabolism in *Trem2*<sup>R47H</sup> rats. Reduction of Aβ42 concentration in the cerebrospinal fluid (CSF) is a biomarker for AD. Aggregation of Aβ42 in brain parenchyma appears the most likely cause for the decreased CSF Aβ42 concentration because the aggregated state inhibits Aβ42 from being transported from the interstitial fluid to the CSF. In absence of aggregation, Aβ40 and Aβ42 CSF levels reflect the concentrations in the CNS. Thus, we measured concentrations of human Aβ38, Aβ40, and Aβ42 in the CSF of peri-adolescent (6–8 weeks old) rats using a human Aβ specific-ELISA, as described previously (*Tambini et al., 2020*). Whereas previously, a decrease in Aβ38 was present in the whole brain lysate of *Trem2*<sup>R47H/R47H</sup> rats, no significant decrease in Aβ38 was seen in the CSF of *Trem2*<sup>R47H</sup> rats (*Figure 1*). No differences were seen in Aβ40 and Aβ42 levels and the Aβ42/Aβ40 ratio between *Trem2*<sup>w/w</sup>, *Trem2*<sup>R47H/w</sup>, and *Trem2*<sup>R47H/R47H</sup> rats (*Figure 1*). Thus, the reduced Aβ clearance caused by the Trem2<sup>R47H</sup> variant in vitro does not result in significant alterations of Aβ steady-state in vivo, at least in young rats.

Next, we looked for other potential early changes in CNS physiology prompted by the *Trem2*<sup>R47H</sup> variant. Macrophages lacking Trem2 produce more TNF-α in response to LPS, zymosan and CpG, suggesting that Trem2 functions to inhibit cytokine production by macrophages (*Turnbull et al., 2006*). Thus, it is possible that the p.R47H variant may alter the anti-inflammatory TREM2 function in microglia. To test for this, we used the Proinflammatory Panel 2 (rat) ELISA multiplex Kit from MSD. This multiplex kit allows for quantitative determination of 9 cytokines (IFN-γ, IL-1β, IL-4, IL-5, IL-6, CXCL1, IL-10, IL-13, and TNF-α) that are important in inflammation response and immune system regulation. First, we measure cytokine levels in the CNS of 4 weeks old rats, the same rats tested previously for Aβ levels (*Tambini and D'Adamio, 2020*). IL-1β, CXCL1 A and TNF-α were

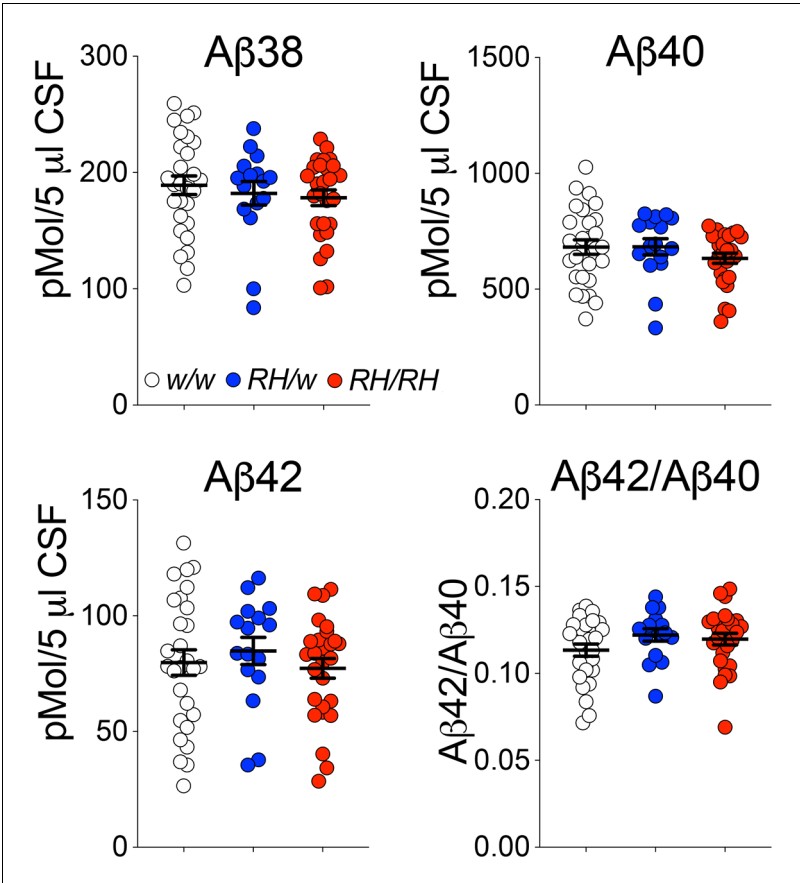

**Figure 1.** Concentrations of human Aβ species are similar in the CSF of peri-adolescent *Trem2*$^{w/w}$, *Trem2*$^{R47H/w}$, and *Trem2*$^{R47H/R47H}$ rats. Levels of Aβ38, Aβ40, and Aβ42/Aβ40 ratio in 6–8 weeks old *Trem2*$^{w/w}$, *Trem2*$^{R47H/w}$, and *Trem2*$^{R47H/R47H}$ rat CSF. We used the following numbers of samples: *Trem2*$^{w/w}$, females n = 12, males n = 16; *Trem2*$^{R47H/w}$ females n = 6, males n = 10; *Trem2*$^{R47H/R47H}$ females n = 10, males n = 17. Data are represented as mean ± SEM. Data were analyzed by ordinary one-way ANOVA. No differences were seen in Aβ38 [$F_{(2, 68)}$=0.5339, p=0.5887], Aβ40 [$F_{(2, 68)}$=1.010, p=0.3696], Aβ42 [$F_{(2, 68)}$=0.4376, p=0.6474] levels and the Aβ42/Aβ40 ratio [$F_{(2, 68)}$=1.564, p=0.2168].

The online version of this article includes the following source data for figure 1:

**Source data 1.** Related to *Figure 1*.

significantly increased in *Trem2*$^{R47H/R47H}$ rats whereas IFN-γ, IL-4, IL-5, IL-6, IL-10 and IL-13 were not (*Figure 2*).

Next, we measure cytokine concentrations in the CSF samples tested for Aβ levels in *Figure 1*. IL-6, IL-10, IL-13 and TNF-α were significantly increased in the CSF of homozygous *Trem2*$^{R47H/R47H}$ rats as compared to *Trem2*$^{w/w}$ controls (*Figure 3*). IL-13 was also significantly higher in heterozygous *Trem2*$^{R47H/w}$ animals, while the increase in TNF-α in these heterozygous animals was close to statistical significance. There are several differences between the cytokine changes observed in the CNS versus the CSF. These differences may be due to several factors, including the different age of the two sets of animals and the fact that distinct cytokines may accumulate differently in the CSF as compared to the CNS. Nevertheless, these results suggest that the Trem2$^{R47H}$ variant increases pro-inflammatory cytokine levels in the CNS of young animals, animals that do not show concurrent changes in human Aβ species concentration. Thus, it is possible that p.R47H pathogenic variant may impact CNS pro-inflammatory cytokines production first and independently of Aβ42 clearance.

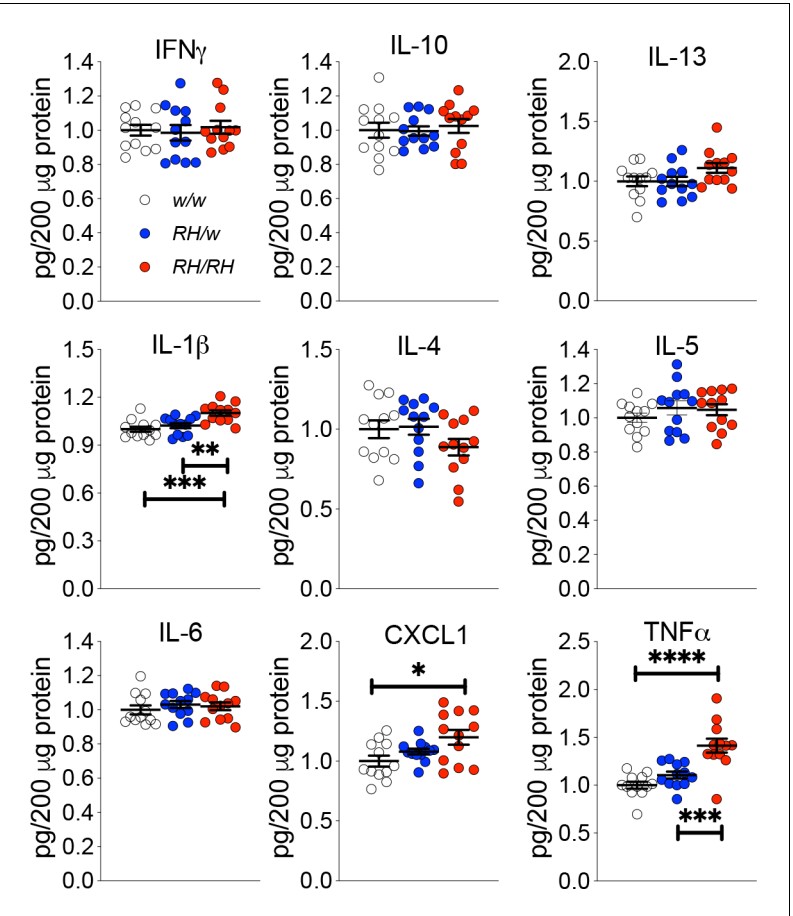

**Figure 2.** Levels of TNF-α and other proinflammatory cytokines are increased in the CNS of pre-adolescent *Trem2^R47H/R47H^* rats. Levels of IFN-δ, IL-1β, IL-4, IL-5, IL-6, CXCL1, IL-10, IL-13, and TNF-α in the CNS of 4 weeks old rats were measured by ELISA. IL-1β, CXCL1 and TNF-α are significantly increased in *Trem2^R47H/R47H^* rats; IFN-γ, IL-4, IL-5, IL-6, IL-10 and IL-13 are not. We used the following numbers of samples: *Trem2^w/w^*, females n = 5, males n = 7; *Trem2^R47H/w^* females n = 6, males n = 6; *Trem2^R47H/R47H^* females n = 6, males n = 6. Data are represented as mean ± SEM and were analyzed by ordinary one-way ANOVA followed by post-hoc Tukey's multiple comparisons test when ANOVA showed significant differences [IFN-γ: $F_{(2, 33)}$=0.1762, p=0.8392 - IL-10: $F_{(2, 33)}$=0.1781, p=0.8376 - IL-13: $F_{(2, 33)}$=2.595, p=0.0898 - IL-1β: $F_{(2, 33)}$=10.52, p=0.0003***; post-hoc Tukey's multiple comparisons test: *w/w vs. RH/w*, p=0.5921 (ns); *w/w vs. RH/RH*, p=0.0003***; *RH/w vs. RH/RH*, p=0.0051** - IL-4: $F_{(2, 33)}$=1.763, p=0.1873 - IL-5: $F_{(2, 33)}$=0.8137, p=0.4519 - IL-6: $F_{(2, 33)}$=0.4640, p=0.6328 - KC-GRO: $F_{(2, 33)}$=4.653, p=0.0166*; post-hoc Tukey's multiple comparisons test: *w/w vs. RH/w*, p=0.4516 (ns); *w/w vs. RH/RH*, p=0.0128*; *RH/w vs. RH/RH*, p=0.1808 (ns) - TNF-α: $F_{(2, 33)}$=17.32, p<0.0001****; post-hoc Tukey's multiple comparisons test: *w/w vs. RH/w*, p=0.3272 (ns); *w/w vs. RH/RH*, p<0.0001****; *RH/w vs. RH/RH*, p=0.0005***].

The online version of this article includes the following source data for figure 2:

**Source data 1.** Related to *Figure 2*.

## *Treml1* mRNA expression is normal in *Trem2^R47H/R47H^* and *Trem2^R47H/w^* rats

Previous studies have shown that *Treml1* could be affected in some *Trem2* knockout models (*Kang et al., 2018*) since the *Treml1* locus is in close proximity with *Trem2*. To test whether the genetic modification used to generate the *Trem2^R47H^* allele altered *Treml1* gene expression we measure the levels of *Treml1* mRNA using qRT-PCR. *Treml1* mRNA levels were measured in both purified microglia (*Figure 4A*) as well as total brain (*Figure 4B*): expression of *Treml1* in *Trem2^R47H/R47H^* and *Trem2^R47H/w^* was comparable to that observed in *Trem2^w/w^* samples. Thus, the introduced *R47H* mutations do not alter *Treml1* mRNA expression. This result is not surprising given that our genetic manipulation results in minimal alteration of the nucleotide sequence of the *Trem2* gene locus (2

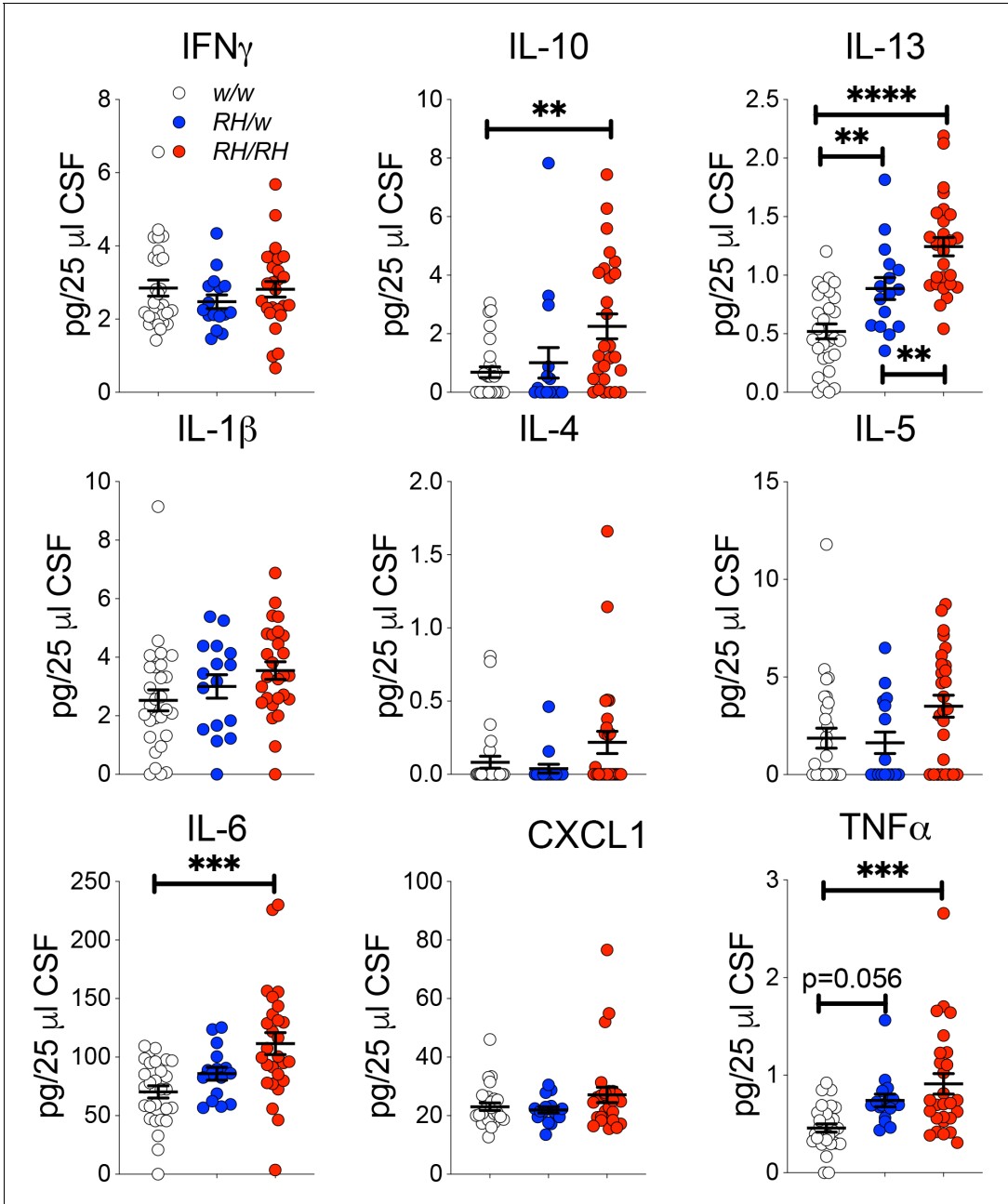

**Figure 3.** Levels of TNF-α and other pro-inflammatory cytokines are increased in the CSF of peri-adolescent *Trem2^R47H/R47H* rats. Measurement of cytokines present in the CSF of 6–8 weeks old rats shows that IL-1β, IL-6, IL-10, IL-13 and TNF-α are significantly increased in *Trem2^R47H/R47H* rats; IFN-δ, IL-1β, IL-4, IL-5 and CXCL1 are not. The same samples used for the experiments shown in *Figure 1* were used here. Data are represented as mean ± SEM and were analyzed by ordinary one-way ANOVA followed by post-hoc Tukey's multiple comparisons test when ANOVA showed significant differences [IFN-γ: $F_{(2, 68)}$=0.7008, p=0.4997- IL-10: $F_{(2, 68)}$=5.651, p=0.0054**; post-hoc Tukey's multiple comparisons test: *w/w vs. RH/w*, p=0.8335 (ns); *w/w vs. RH/RH*, p=0.0051**; *RH/w vs. RH/RH*, p=0.0779 (ns) - ANOVA summary of IL-13: $F_{(2, 68)}$=26.21, p<0.0001****; post-hoc Tukey's multiple comparisons test: *w/w vs. RH/w*, p=0.0066**; *w/w vs. RH/RH*, p<0.0001****; *RH/w vs. RH/RH*, p=0.0090** - IL-1β: $F_{(2, 68)}$=2.473, p=0.0919 - IL-4: $F_{(2, 68)}$=2.504, p=0.0893 - IL-5: $F_{(2, 68)}$=3.489, p=0.0361*; post-hoc Tukey's multiple comparisons test: *w/w vs. RH/w*, p=0.9545 (ns); *w/w vs. RH/RH*, p=0.069 (ns); *RH/w vs. RH/RH*, p=0.0753 (ns) - IL-6: $F_{(2, 68)}$=9.016, p=0.0003***; post-hoc Tukey's multiple comparisons test: *w/w vs. RH/w*, p=0.361 (ns); *w/w vs. RH/RH*, p=0.0002***; *RH/w vs. RH/RH*, p=0.0707 (ns) – CXCL1: $F_{(2, 68)}$=1.847, p=0.1656 - TNF-α: $F_{(2, 68)}$=9.720, p=0.0002***; post-hoc Tukey's multiple comparisons test: *w/w vs. RH/w*, p=0.0565 (ns); *w/w vs. RH/RH*, p=0.0001***; *RH/w vs. RH/RH*, p=0.3403 (ns)].

The online version of this article includes the following source data for figure 3:

**Source data 1.** Related to *Figure 3*.

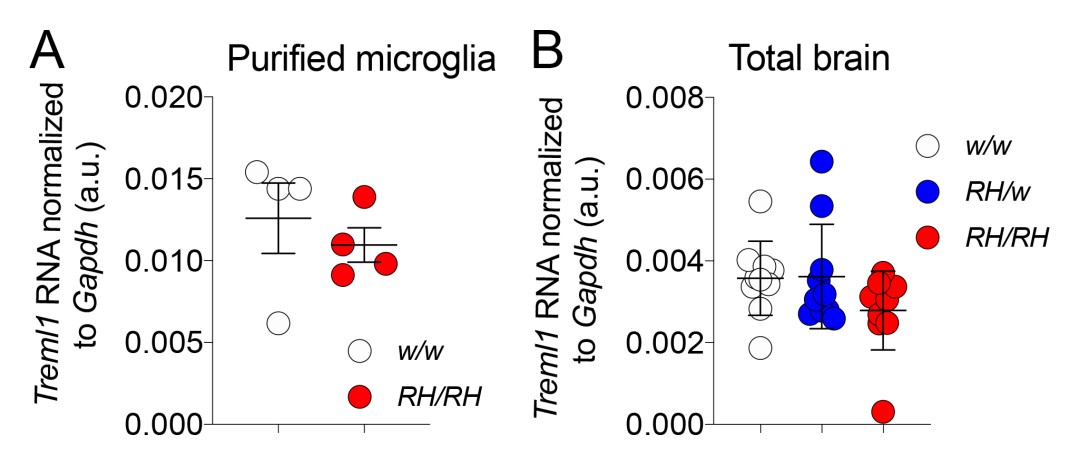

**Figure 4.** *Treml1* mRNA expression is normal in *Trem2^{R47H}* rats. (**A**) Levels of *Treml1* mRNA were measured and normalized to *Gapdh* mRNA expression. We used microglia purified from 2 male and 2 female 5/6 weeks old rats for each genotype (*Trem2^{w/w}* and *Trem2^{R47H/R47H}*). Data were analyzed by unpaired student's t-test (p=0.5198), and presented as average (*Treml1/Gapdh*)± SEM. (**B**) *Treml1* mRNA expression was measured in total brains. We used 5 female and 5 male (6–8 weeks of age) for each genotype (*Trem2^{w/w}*, *Trem2^{R47H/w}* and *Trem2^{R47H/R47H}*). Data are represented as (*Treml1/Gapdh*) mean ± SEM and were analyzed by ordinary one-way ANOVA ($F_{(2, 27)}$=1.940, p=0.1632).

The online version of this article includes the following source data for figure 4:

**Source data 1.** Related to *Figure 4A,B*.

bases) with no deletions. In contrast, the *Trem2* knockout mouse model showing altered *Treml1* gene expression was generated from the Velocigene 'definitive null' targeting strategy. In this strategy the entire coding region was replaced by selection cassette lacZ-flox-human Ubiquitin C promoter-neomycin-flox, beginning at 16 bp upstream of the ATG start codon and ending at the TGA stop codon of mouse *Trem2*.

## Augmented excitatory synaptic transmission at hippocampal SC–CA3 >CA1 synapses of peri-adolescent rats carrying the *Trem2^{R47H}* variant

Among the cytokines tested, TNF-α is the only one that is significantly increased in both pre-adolescent CNS tissue and peri-adolescent CSF. Physiological levels of TNF-α produced by glia cells are required for normal surface expression of AMPA (α-amino-3-hydroxy-5-methyl-4-isoxazole propionic acid) receptors at synapses, and increased TNF-α concentrations cause a rapid exocytosis of AMPA receptors in hippocampal pyramidal neurons increasing excitatory synaptic strength (*Beattie et al., 2002*; *Ogoshi et al., 2005*; *Stellwagen et al., 2005*; *Stellwagen and Malenka, 2006*). Interestingly, *Amyloid Precursor Protein* (*APP*, which codes for the precursor of Aβ) (*Del Prete et al., 2014*; *Fanutza et al., 2015*; *Groemer et al., 2011*; *Kohli et al., 2012*; *Lundgren et al., 2015*; *Norstrom et al., 2010*; *Tambini et al., 2019*; *Yao et al., 2019a*), *Presenilin 1/2* (*PSEN1* and *PSEN2*, which code for the catalytic components of the γ-secretase complex, an enzyme essential for Aβ production) (*Wu et al., 2013*; *Xia et al., 2015*) and *Integral Membrane Protein 2B* (*ITM2b*, whose protein product binds APP and regulates APP processing) (*Fotinopoulou et al., 2005*; *Matsuda et al., 2005*; *Matsuda et al., 2008*; *Matsuda et al., 2011*; *Yao et al., 2019b*) play a physiological role in glutamatergic transmission. Moreover, some *APP*, *PSEN1* and *ITM2b* mutations linked to familial dementia alter this physiological function (*Tamayev et al., 2010a*; *Tamayev et al., 2010b*; *Tambini et al., 2019*; *Xia et al., 2015*; *Yao et al., 2019b*). Based on these data, we examined the effects of the *Trem2^{R47H}* variant on glutamatergic synaptic transmission in the hippocampal Schaffer-collateral pathway. First, we analyzed miniature excitatory postsynaptic currents (mEPSC). The amplitude of mEPSC, which is dependent on postsynaptic AMPA receptors, was significantly increased in *Trem2^{R47H/R47H}* rats (*Figure 5B,E and F*) while the decay time was unchanged (*Figure 5C and E*). In addition, the frequency of mEPSC was significantly increased in *Trem2^{R47H/R47H}* rats (*Figure 5D and G*). To test further whether postsynaptic AMPAR-mediated responses are increased in *Trem2^{R47H}*

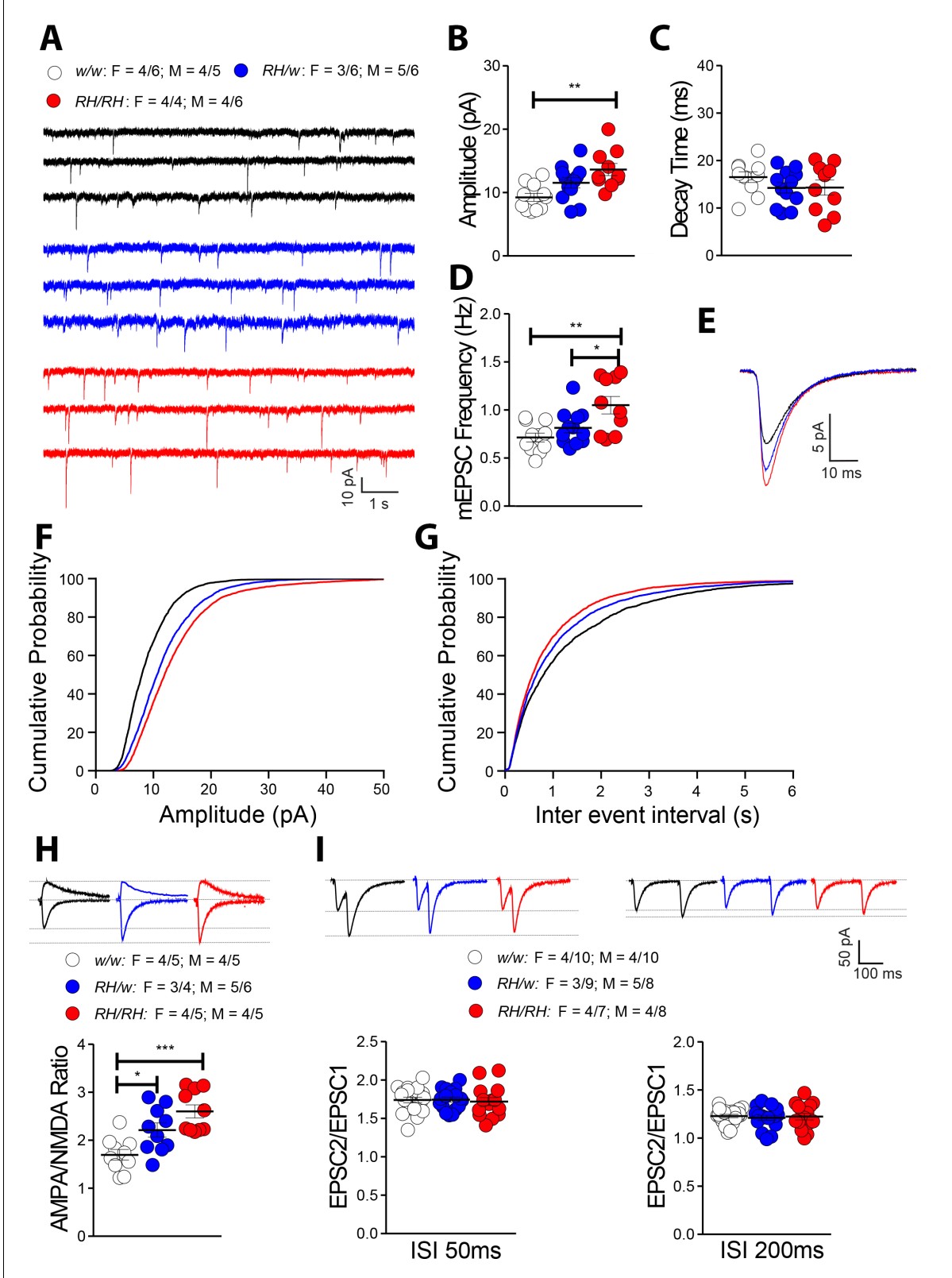

**Figure 5.** The *Trem2R47H* variant alters glutamatergic synaptic transmission in a gene dosage-dependent manner in young rats. (**A**) Representative recording traces of mEPSC at SC–CA3 >CA1 pyramidal cell synapses. (**B**) The *Trem2R47H* variant causes a significant increase in mEPSC amplitude [$F_{(2, 30)}$=7.371, p=0.0025**; post-hoc Tukey's multiple comparisons test: *w/w vs. RH/w*, p=0.1093 (ns); *w/w vs. RH/RH*, p=0.0017**; *RH/w vs. RH/RH*, p=0.1630 (ns)]. (**C**) In contrast, decay time of mEPSC was not changed [$F_{(2, 30)}$=1.396, p=0.2632]. (**D**) Frequency of mEPSC was enhanced by the *Trem2R47H* variant

Figure 5 continued

[$F_{(2, 30)}$=7.092, p=0.0030**; post-hoc Tukey's multiple comparisons test: *w/w* vs. *RH/w*, p=0.4881 (ns); *w/w* vs. *RH/RH*, p=0.0025**; *RH/w* vs. *RH/RH*, p=0.0345*]. (E) Average mEPSC of the three groups depicts differences in amplitude. As can also be noted in B, $Trem2^{R47H/w}$ rats show mEPSC with increased amplitude, albeit this increase did not reach statistical significance. Cumulative probability of AMPAR-mediated mEPSC amplitudes (F) and inter event intervals (G). (H) AMPA/NMDA ratio is significantly increased in both $Trem2^{R47H/R47H}$ and $Trem2^{R47H/w}$ rats in a gene dosage dependent manner [$F_{(2, 27)}$=11.75, p=0.0002***; post-hoc Tukey's multiple comparisons test: *w/w* vs. *RH/w*, p=0.0265*; *w/w* vs. *RH/RH*, p=0.0001***; *RH/w* vs. *RH/RH*, p=0.1160 (ns)]. Representative traces are shown on of the graph (traces are averaged from 20 sweeps). (I) Average PPF at 50 ms (left panel) and 200 ms (right panel) Inter stimulus Interval (ISI) [PPF at 50 ms ISI: $F_{(2, 49)}$=0.0949, p=0.9096; PPF at 200 ms ISI: $F_{(2, 49)}$=0.1397, p=0.8700]. Representative traces are shown on top of the panels. Data are represented as mean ± SEM and were analyzed by ordinary one-way ANOVA followed by post-hoc Tukey's multiple comparisons test when ANOVA showed significant differences. For each type of recordings, we indicate the number of animals by genotype and sex, plus the number of recording by genotype and sex as follow: 1) genotypes: *w/w* = Trem2 w/w, *RH/w* = Trem2 R47H/w, *RH/RH* = $Trem2^{R47H/R47H}$;2) sex: R47H/R47H = female, M = males; 3) number of animals and number of recordings from animals: n/n', were n = number of animals, n'=number of recordings from the n animals. For example, the *w/w*: F = 4/6; M = 4/5 in A indicates that data for mEPSC for the $Trem2^{w/w}$ rats were obtained from 4 females and 4 males, and that 6 recordings were obtained from the 4 females and 5 recordings from the 4 males.

The online version of this article includes the following source data for figure 5:

**Source data 1.** Related to *Figure 5B,C,D,H,I*.

---

rats, we measured AMPAR and NMDAR-dependent synaptic responses. Consistent with the hypothesis that the $Trem2^{R47H}$ variant increases AMPAR-mediated responses, the AMPA/NMDA ratio was increased in both $Trem2^{R47H/R47H}$ and $Trem2^{R47H/w}$ rats and the significance of the increase is gene dosage-dependent (*Figure 5H*).

Treatment of primary neurons with recombinant TNF-α (*Beattie et al., 2002*; *Grassi et al., 1994*) and *Trem2* deletion in mice (*Filipello et al., 2018*) enhance frequency of mEPSC. Several mechanisms, including an increase in release Probability (P*r*) of glutamatergic synaptic vesicles and/or synaptic density, can enhance frequency of mEPSC. Paired-pulse facilitation (PPF), a form of short-term synaptic plasticity, is determined, at least in part, by changes in P*r*; an increase in P*r* leads to a decrease in facilitation (*Zucker and Regehr, 2002*). PPF was not significantly changed in $Trem2^{R47H/w}$ and $Trem2^{R47H/R47H}$ rats as compared to $Trem2^{w/w}$ animals (*Figure 5I*). This evidence argue against changes in P*r* in $Trem2^{R47H/R47H}$ rats and suggests that enhanced mEPSC frequency may be due to increased synaptic density, a phenomenon that in $Trem2^{-/-}$ mice is due to impaired synapse elimination by microglia (*Filipello et al., 2018*).

## Supraphysiological TNF-α concentrations boost glutamatergic transmission in hippocampal SC–CA3 >CA1 synapses of young $Trem2^{R47H/R47H}$ rats

The similarities between glutamatergic transmission alterations observed in young $Trem2^{R47H}$ rats and those induced by TNF-α are striking. To test whether the supraphysiological TNF-α concentrations trigger glutamatergic deficits in $Trem2^{R47H/R47H}$ rats, we treated hippocampal slices with a neutralizing antibody to rat TNF-α (anti-TNF-α), which functions as a TNF-α antagonist. To control for off-target effects of the antibody, we used a Goat IgG isotype control. The 50% neutralization dose ($ND_{50}$) of this anti-TNF-α antibody against the cytotoxic effect of recombinant rat TNF-α (0.25 ng/mL) is about 500 ng/ml. Since physiological levels of TNF-α are necessary for normal glutamatergic transmission and most of the activities of TNF-α can be rapidly reversed (*Beattie et al., 2002*; *Ogoshi et al., 2005*; *Stellwagen et al., 2005*; *Stellwagen and Malenka, 2006*), we tested the acute effects 10 ng/ml of anti-TNF-α, a concentration ~50 times lower than the $ND_{50}$. At this concentration, anti-TNF-α occluded the increase in mEPSC amplitude (*Figure 6A,B,E and F*) and AMPA/NMDA ratio (*Figure 6H*). Anti-TNF-α did not alter decay time of mEPSC (*Figure 6C*) and PPF (*Figure 6I*), which were not significantly changed in $Trem2^{R47H/R47H}$ rats (see *Figure 5C and I*). Surprisingly, anti-TNF-α did not reduce mEPSC frequency of $Trem2^{R47H/R47H}$ SC–CA3 >CA1 synapses to WT-like levels (*Figure 6D and F*).

The evidence that the goat IgG isotype control did not occlude glutamate transmission alterations observed in the mutant rats (*Figure 6*) indicates that the effects of anti-TNF-α are specific. The finding that these low doses of anti-TNF-α do not alter glutamatergic transmission in $Trem2^{w/w}$ rats (*Figure 6*) demonstrates that, at least at this dosage, anti-TNF-α only targets synaptic transmission alterations triggered by excess TNF-α set off by the $Trem2^{R47H}$ variant. Overall, these data indicate

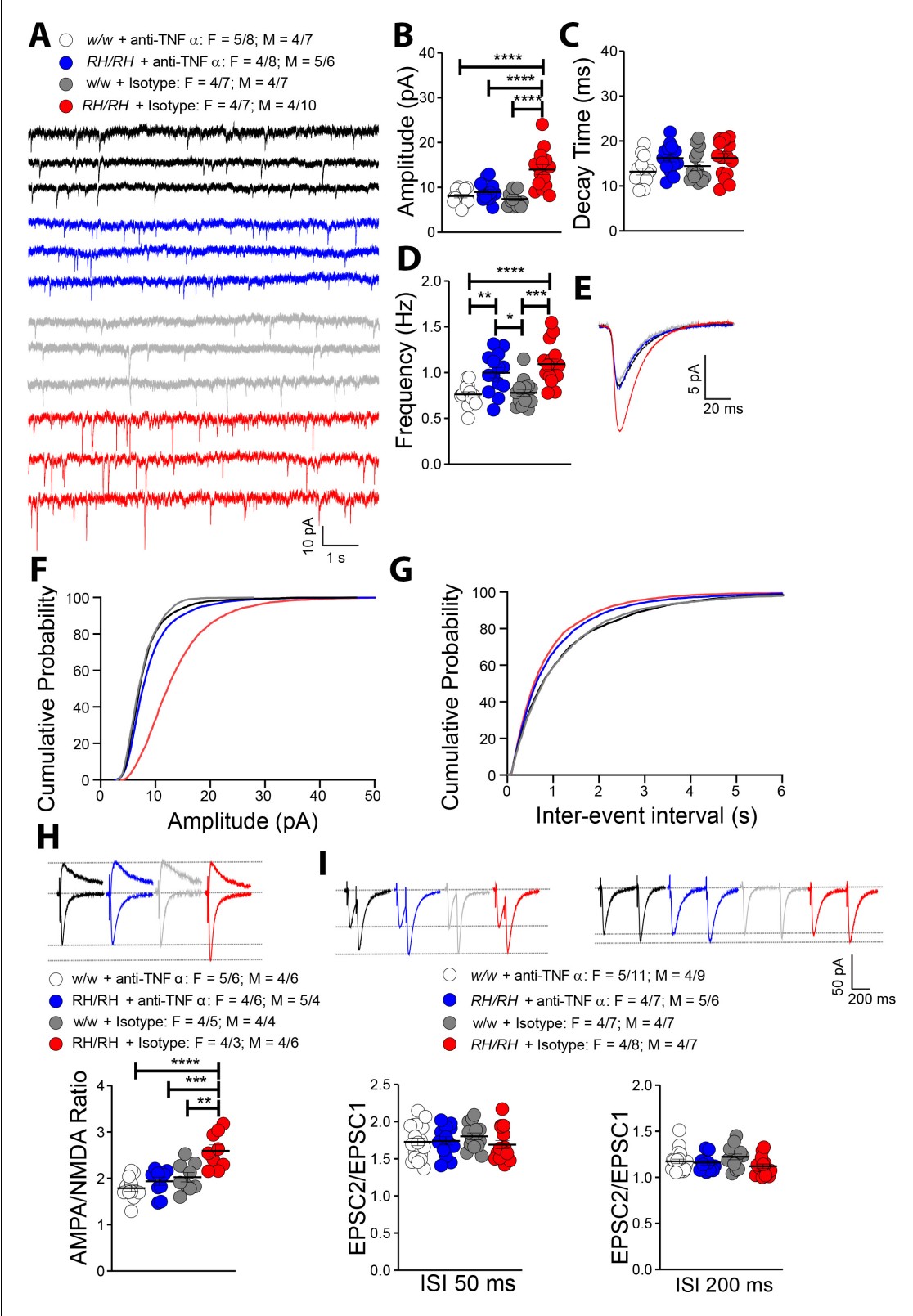

**Figure 6.** Rapid reduction of excess TNF-α activity normalizes amplitude of glutamatergic synaptic responses in young *Trem2*[R47H/R47H] rats. (**A**) Representative mEPSC traces. (**B**) The increase in mEPSC amplitude caused by the *Trem2*[R47H] variant is occluded by anti-TNF-α application but not by the IgG isotype control [$F_{(3, 54)}=18.79$, p<0.0001****; post-hoc Tukey's multiple comparisons test: *w/w* + anti-TNF-α *vs. RH/RH* + anti-TNF-α, p=0.7884 (ns); *w/w* + anti-TNF-α *vs. RH/RH* + Isotype, p<0.0001****; *w/w* + anti-TNF-α *vs. w/w* + Isotype, p=0.9252 (ns); *RH/RH* + anti-TNF-α *vs. RH/RH* +

*Figure 6 continued on next page*

Figure 6 continued

Isotype, p<0.0001****; RH/RH + anti-TNF-α vs. w/w + Isotype p=0.4299 (ns); RH/RH + Isotype vs. w/w + Isotype, p<0.0001****]. (C) Decay time of mEPSC was not changed by either genotype or treatments [$F_{(3, 54)}$=2.716, p=0.0536]. (D) The increased frequency of mEPSC observed in $Trem2^{R47H/R47H}$ rats was not significantly affected by either anti-TNF-α or isotype control IgG [$F_{(3, 54)}$=12.05, p<0.0001****; post-hoc Tukey's multiple comparisons test: w/w + anti-TNF-α vs. RH/RH + anti-TNF-α, p=0.0046**; w/w + anti-TNF-α vs. RH/RH + Isotype, p<0.0001****; w/w + anti-TNF-α vs. w/w + Isotype, p=0.9947 (ns); RH/RH + anti-TNF-α vs. RH/RH + Isotype, p=0.5147 (ns); RH/RH + anti-TNF-α vs. w/w + Isotype p=0.0109*; RH/RH + Isotype vs. w/w + Isotype, p<0.0001****]. (E) Average mEPSC of the four groups. Cumulative probability of AMPAR-mediated mEPSC amplitudes (F) and inter event intervals (G). (H) The increase of AMPA/NMDA ratio observed in $Trem2^{R47H/R47H}$ rats was reversed by anti-TNF-α but not IgG isotype control [$F_{(3, 36)}$=13.50, p<0.0001****; post-hoc Tukey's multiple comparisons test: w/w + anti-TNF-α vs. RH/RH + anti-TNF-α, p=0.6507 (ns); w/w + anti-TNF-α vs. RH/RH + Isotype, p<0.0001****; w/w + anti-TNF-α vs. w/w + Isotype, p=0.2880 (ns); RH/RH + anti-TNF-α vs. RH/RH + Isotype, p=0.0002***; RH/RH + anti-TNF-α vs. w/w + Isotype p=0.9171 (ns); RH/RH + Isotype vs. w/w + Isotype, p<0.0015**]. Representative traces are shown above the graph (traces are averaged from 20 sweeps). (I) Neither genotype nor treatment changed average PPF at 50 ms (left panel) and 200 ms (right panel) ISI [PPF at 50 ms ISI: $F_{(3, 58)}$=0.7420, p=0.5313; PPF at 200 ms ISI: $F_{(3, 58)}$=2.356, p=0.0812]. Representative traces are shown above the panels. Data are represented as mean ± SEM and were analyzed by ordinary one-way ANOVA followed by post-hoc Tukey's multiple comparisons test when ANOVA showed significant differences. Number of animals and of recordings are shown as explained in **Figure 5**.

The online version of this article includes the following source data for figure 6:

**Source data 1.** Related to *Figure 6B,C,D,H,I*.

that the increase in amplitude of AMPAR-mediated responses at SC–CA3 >CA1 synapses of $Trem2^{R47H/R47H}$ rats is due to the acute action of supraphysiological TNF-α concentrations prompted by the $Trem2^{R47H}$ variant and is rapidly reversible.

## LTP is suppressed at hippocampal SC–CA3 >CA1 synapses of young $Trem2^{R47H/R47H}$ rats: this deficit is rescued by reducing TNF-α activity

LTP, a long-lasting form of synaptic plasticity, has been described at glutamatergic synapses throughout the brain and remains one of the most attractive cellular models for learning and memory. Given the alterations in glutamatergic transmission observed in $Trem2^{R47H}$ rats, we determined whether the $Trem2^{R47H}$ variant could also impact this electrophysiological surrogate of memory in young rats (6–8 weeks old peri-adolescent rats). Before recording LTP, baseline was recorded every minute at an intensity that elicited a response 40% of the maximum evoked response. Maximum evoked responses were assessed by measuring the slope of the field excitatory postsynaptic potential (fEPSP) elicited by stimuli of increasing intensity (basal synaptic transmission or BST). Consistent with the evidence that $Trem2^{R47H}$ variant enhances AMPAR-mediated responses, BST was increased in $Trem2^{R47H/R47H}$ rats (**Figure 7A**). Surprisingly perhaps given the young age of the animals, LTP was reduced in $Trem2^{R47H/R47H}$ rats (**Figure 7B and C**), indicating that the $Trem2^{R47H}$ variant compromises LTP early in life. Interestingly, anti-TNF-α occluded the increase in BST (**Figure 7D**) and suppression of LTP (**Figure 7E and F**). This effect is specific and indicates that also LTP suppression in young $Trem2^{R47H/R47H}$ rats is caused by the sustained and acute action of supraphysiological TNF-α concentrations prompted by the $Trem2^{R47H}$ variant and is rapidly reversible.

## Discussion

Here, we studied young rats carrying the $Trem2^{R47H}$ variant, which increases the risk for AD in humans, with the purpose of probing early dysfunctions that may underlie initial pathogenic mechanisms leading to dementia. We found no alteration in Aβ metabolism (**Figure 1**); however, pro-inflammatory cytokines, especially TNF-α, were significantly increased in the CNS and CSF of rats carrying $Trem2^{R47H}$ variant. Overall, these increases were more significant in $Trem2^{R47H/R47H}$ as opposed to $Trem2^{R47H/w}$ rats (**Figures 2** and **3**), suggesting that early neuroinflammation caused by the $Trem2^{R47H}$ variant is gene dosage-dependent. In the CNS, *TREM2* expression is restricted to microglia (**Schmid et al., 2002**) and activated microglia secrete pro-inflammatory cytokines: thus, increased secretion of pro-inflammatory cytokines by $Trem2^{R47H}$-microglia may cause early neuroinflammation. Also, microglia may promote cytokine production by other CNS cell types -such as astrocytes, for example- and/or that excess cytokines present in the brain of $Trem2^{R47H}$ rats may be produced by non-CNS resident cells. It is also possible that cytokine clearance is altered in $Trem2^{R47H}$ rats. These possibilities do not need to be mutually exclusive.

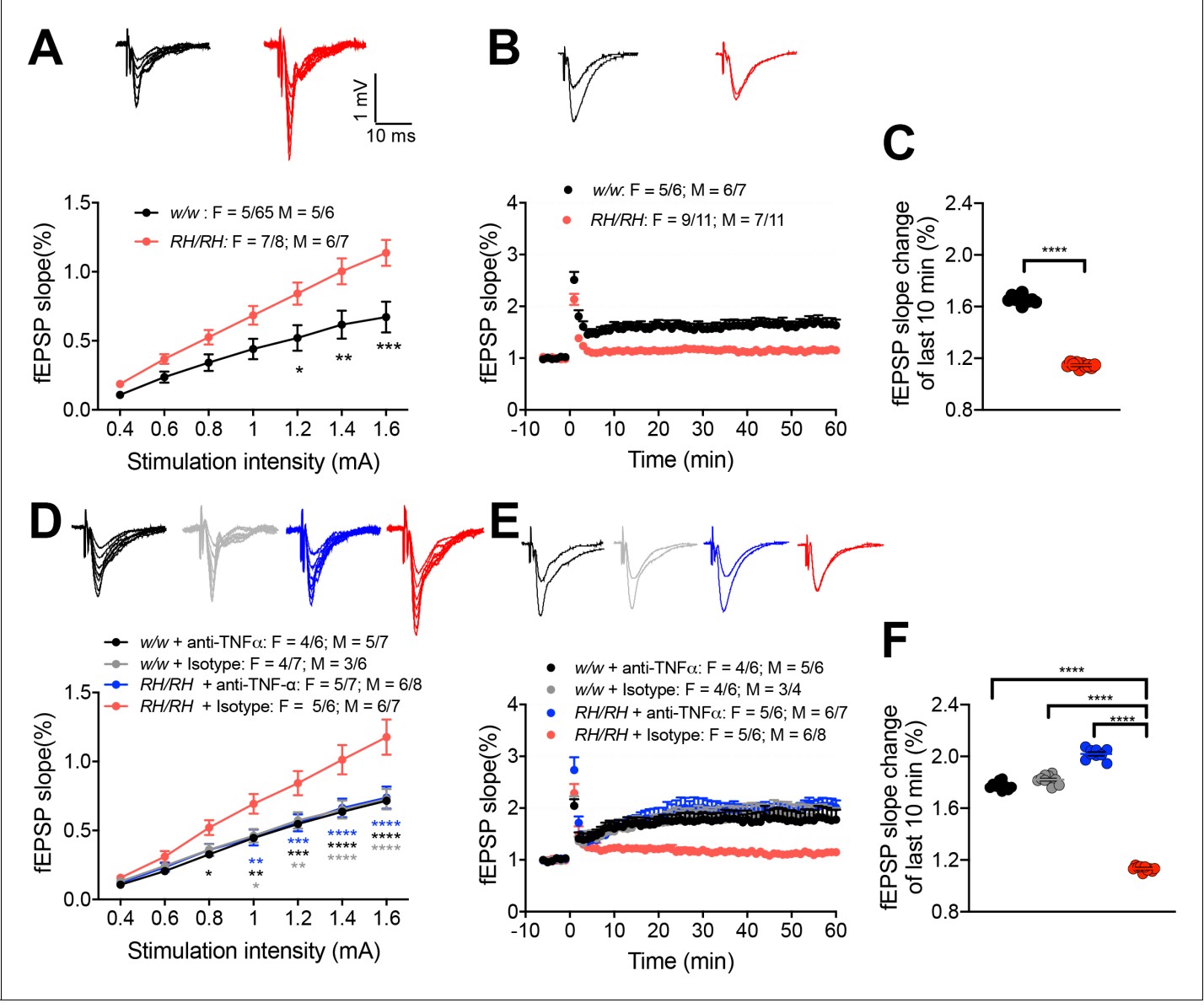

**Figure 7.** LTP is impaired in RH mutant rats, and the impairment could be rescued by application of anti-TNF-α antibody. (A) The input-output slope is significantly increased in $Trem2^{R47H/R47H}$ rats [two-way ANOVA, stimulation intensity x genotype interaction $F_{(6, 144)}=6.745$, p<0.0001****; post-hoc Sidak's multiple comparisons test: 1.2 mV p=0.0175*; 1.4 mV: p=0.0021**; 1.6 mV: p=0.0001***] Representative traces of fEPSPs in response to increasing stimulus from 0.4 to 1.6 mA are shown on the top. (B) LTP is impaired in $Trem2^{R47H/R47H}$ rats. Average traces of the baseline and the last 5mins of LTP are shown on top. (C) Plot of fEPSP slope change of the last 10 min of LTP in B (unpaired t test, p<0.0001****). (D) The increase of input-out slope in $Trem2^{R47H/R47H}$ rats is reversed by application of anti-TNF-α [ANOVA for repeated measures $F_{(18, 300)}=6.579$, p<0.0001****; post-hoc Tukey's multiple comparisons test: 0.8 mV RH/RH + Isotype vs. w/w + anti-TNF-α p=0.0479*; 1.0 mV RH/RH + Isotype vs. RH/RH + anti-TNF-α p=0.0061**, RH/RH + Isotype vs. w/w + anti-TNFα p=0.0072**, RH/RH + Isotype vs. w/w + Isotype p=0.0113*; 1.2 mV RH/RH + Isotype vs. RH/RH + anti-TNFα p=0.0009***, RH/RH + Isotype vs. w/w + anti-TNF-α p=0.0009***, RH/RH + Isotype vs. w/w + Isotype p=0.0029**; 1.4 mV RH/RH + Isotype vs. RH/RH + anti-TNF-α p<0.0001****, RH/RH + Isotype vs. w/w + anti-TNF-α p<0.0001****, RH/RH + Isotype vs. w/w + Isotype p=P < 0.0001****; 1.6 mV RH/RH + Isotype vs. RH/RH + anti-TNF-α p<0.0001****, RH/RH + Isotype vs. w/w + anti-TNF-α P p<0.0001****, RH/RH + Isotype vs. w/w + Isotype p<0.0001****]. Representative fEPSP traces are shown on top. (E) The impaired LTP is restored by application of anti-TNF-α antibody. The average traces of the baseline and the last 5mins of LTP are shown on top. (F) Plot of fEPSP slope change of the last 10 min of LTP in E [one-way ANOVA, $F_{(3, 36)}=1490$, p<0.0001; post-hoc Tukey's multiple comparisons test: RH/RH + Isotype vs. RH/RH + anti-TNF-α p<0.0001****, RH/RH + Isotype vs. w/w + anti-TNF-α P p<0.0001****, RH/RH + Isotype vs. w/w + Isotype p<0.0001****].

The online version of this article includes the following source data for figure 7:

**Source data 1.** Related to *Figure 7A,C,D,F*.

Three lines of evidence prompted us to test whether TNF-α, which was significantly increased in both pre-adolescent CNS tissue and peri-adolescent CSF (*Figures 2* and *3*), could affect glutamatergic synaptic transmission in *Trem2^R47H^* rats: 1) TNF-α produced by glia cells is required for normal surface expression of AMPA receptors at synapses and increased TNF-α concentrations cause a rapid exocytosis of AMPAR in hippocampal pyramidal neurons increasing excitatory synaptic strength *Beattie et al., 2002*; *Ogoshi et al., 2005*; *Stellwagen et al., 2005*; *Stellwagen and Malenka, 2006*; 2) several genes linked to dementia, including *APP*, *PSEN1*, *PSEN2* and *ITM2b* play a physiological role in glutamatergic transmission (*Del Prete et al., 2014*; *Fanutza et al., 2015*; *Fotinopoulou et al., 2005*; *Groemer et al., 2011*; *Kohli et al., 2012*; *Lundgren et al., 2015*; *Matsuda et al., 2005*; *Matsuda et al., 2008*; *Matsuda et al., 2011*; *Norstrom et al., 2010*; *Tambini et al., 2019*; *Wu et al., 2013*; *Xia et al., 2015*; *Yao et al., 2019a*; *Yao et al., 2019b*); 3) *APP*, *PSEN1* and *ITM2b* mutations linked to familial dementia alter this physiological functions (*Tamayev et al., 2010a*; *Tamayev et al., 2010b*; *Tambini et al., 2019*; *Xia et al., 2015*; *Yao et al., 2019b*). Analysis of excitatory synaptic transmission at hippocampal SC–CA3 >CA1 synapses of young animals showed that the *Trem2^R47H^* variant increases, in a gene dosage-dependent fashion, AMPAR-mediated glutamatergic responses (*Figure 5B,E,F and I*; *Figure 7A*). Consistent with the hypothesis, a neutralizing anti-TNF-α antibody, which functions as a TNF-α antagonist, occludes these alterations (*Figure 6B,E,F and I*; *Figure 7C*). Thus, the increase in amplitude of AMPAR-mediated responses at SC–CA3 >CA1 synapses observed in young *Trem2^R47H/R47H^* rats is due to supraphysiological TNF-α concentrations caused by the *Trem2^R47H^* variant. These alterations must be due to an acute and persistent action of supraphysiological TNF-α concentrations because they are rapidly reversed by low doses of neutralizing anti-TNF-α. Whether these effects are mediated by the known role of TNF-α on AMPAR trafficking at synapses (*Beattie et al., 2002*; *Ogoshi et al., 2005*; *Stellwagen et al., 2005*; *Stellwagen and Malenka, 2006*) remains to be determined.

The frequency of mEPSC is significantly increased in *Trem2^R47H/R47H^* rats as well (*Figure 5D and G*) independently of changes in P$r$ of glutamatergic synaptic vesicles (*Figure 5I*). However short-term treatment with anti-TNF-α did not completely occlude this increase (*Figure 6D and G*). Thus, either the increased mEPSC frequency in *Trem2^R47H/R47H^* rats is TNF-α independent or TNF-α underlies this alteration *via* long-lasting mechanisms that are not fully occluded by short-term treatments with anti-TNF-α. The evidence that recombinant TNF-α enhances mEPSC frequency in primary neurons (*Beattie et al., 2002*; *Grassi et al., 1994*) and that anti-TNF-α reduces the statistical significance of the differences between *Trem2^R47H/R47H^* and *Trem2^w/w^* recordings (w/w + anti-TNF-α vs. RH/RH + anti-TNF-α, p=0.0046**; w/w + anti-TNF-α vs. RH/RH + Isotype, p<0.0001****; w/w + Isotype vs RH/RH + anti-TNF-α. p=0.0109*; w/w + Isotype vs RH/RH + Isotype, p<0.0001****) would support the latter possibility. Increase in synaptic density, a phenomenon that in *Trem2^-/-^* mice is due to impaired synapse elimination by microglia (*Filipello et al., 2018*), can also enhance frequency of mEPSC. Further studies will be needed to determine whether the *Trem2^R47H^* variant operates with a similar mechanism and to determine the role(s) of other pro-inflammatory cytokines whose brain concentration is increased by the *Trem2^R47H^* variant.

Supraphysiological TNF-α concentrations also cause LTP suppression at hippocampal SC–CA3 >CA1 synapses of young *Trem2^R47H/R47H^* rats. Given that LTP is the most attractive cellular model for learning and memory and is considered an electrophysiological surrogate of memory (*Nicoll, 2017*), it would not be surprising if subsequent studies will find that rats carrying the *Trem2^R47H^* variant develop learning and memory deficits. If this were the case, this model organism may be used to test therapeutic approaches, including reduction of TNF-α activity and glutamatergic transmission, targeting early pathogenic mechanisms leading to dementia.

In conclusion, we present evidence that the *Trem2^R47H^* KI rat represents a useful model to dissect early events leading to sporadic forms of dementia. Our study indicates that the microglia-specific pathogenic *Trem2^R47H^* variant boosts glutamatergic neuronal transmission and suppresses LTP by enhancing brain TNF-α concentrations, directly linking microglial to neuronal dysfunctions. More studies will be needed to determine whether this microglia-neuronal axis represents an early, Aβ-independent pathway that facilitates dementia pathogenesis in humans.

# Materials and methods

## Key resources table

| Reagent type (species) or resource | Designation | Source or reference | Identifiers | Additional information |
|---|---|---|---|---|
| Genetic reagent (*Rattus Norvegicus*) | *App*[h] | **Tambini et al., 2019.** Aging Cell 18: e13033 | | Rat *App* allele with humanize Aβ region |
| Genetic reagent (*Rattus Norvegicus*) | *Trem2*[R47H] | **Tambini and D'Adamio, 2020** Sci Rep 10: 4122 | | Rat *Trem2* allele with R47H mutation |
| Commercial assay or kit | V-PLEX Plus Aβ Peptide Panel 1 | Meso Scale Discovery | Cat# K15200G | Used following manufacturer's recommendations |
| Commercial assay or kit | V-PLEX Proinflammatory Panel 2 | Meso Scale Discovery | Cat# K15059D | Used following manufacturer's recommendations |
| Commercial assay or kit | CD11b/c (Microglia) Micro-Beads, rat antibody Cat# 130-105-634 | Miltenyi Biotec | RRID:AB_2783886 | Used following manufacturer's recommendations |
| Commercial assay or kit | Adult Brain Dissociation Kit | Miltenyi Biotec | Cat# 130-107-677 | Used following manufacturer's recommendations |
| Commercial assay or kit | RNeasy RNA Isolation kit | Qiagen | Cat# 74106 | Used following manufacturer's recommendations |
| Commercial assay or kit | High-Capacity cDNA RT kit | Thermo | Cat# 4368814) | Used following manufacturer's recommendations |
| Commercial assay or kit | TaqMan Fast Advanced Mix | Thermo | Cat# 4444556 | Used following manufacturer's recommendations |
| Commercial assay or kit | *Gapdh* Real-Time PCR | Thermo | Rn01775763_g1 | Used following manufacturer's recommendations |
| Commercial assay or kit | *Treml1* Real-Time PCR | Thermo | Rn01511908_g1 | Used following manufacturer's recommendations |
| Antibody | Polyclonal Goat IgG anti-Rat TNFα Cat# AF-510-NA | R and D Systems | RRID:AB_354511 | 10 ng/ml in ACSF |
| Antibody | Polyclonal Goat IgG. antibody Cat# AB-108-C | R and D Systems | RRID:AB_354267 | 10 ng/ml in ACSF |
| Software, algorithm | LinRegPCR software | hartfaalcentrum.nl | | |
| Software, algorithm | pCLAMP10 software | Molecular Devices, | | |
| Software, algorithm | Image Lab software | Biorad | RRID:SCR_014210 | |
| Software, algorithm | GraphPad Prism | | RRID:SCR_002798 | |

## Rat brain preparation

Rats were anesthetized with isoflurane and perfused via intracardiac catheterization with ice-cold PBS. This perfusion step eliminates cytokines and Aβ present in blood. Brains were extracted and homogenized using a glass-teflon homogenizer (w/v = 100 mg tissue/1 ml buffer) in 250 mM Sucrose, 20 mM Tris-base pH 7.4, 1 mM EDTA, 1 mM EGTA plus protease and phosphatase inhibitors (ThermoScientific), with all steps carried out on ice or at 4°C. Total lysate was solubilized with 0.1% SDS and 1% NP-40 for 30 min rotating. Solubilized lysate was spun at 20,000 g for 10 min, the supernatant was collected and analyzed by ELISA.

## Elisa

Aβ38, Aβ40, and Aβ42 were measured with V-PLEX Plus Aβ Peptide Panel 1 6E10 (K15200G) and V-PLEX Plus Aβ Peptide Panel 1. Cytokines (IFN-γ, IL-1β, IL-4, IL-5, IL-6, IL-10, IL-13, CXCL1, and TNF-α) were measured with V-PLEX Proinflammatory Panel 2 Rat Kit (K15059D). Measurements were performed according to the manufacturer's recommendations. Plates were read on a MESO Quick-Plex SQ 120. Data were analyzed using Prism software and represented as mean ± SEM.

## Rat brain preparation for RNA extraction

6–8 week old rats were perfused with PBS (N = 5 for each genotype (*RH/RH*, *RH/w* and *w/w*) and each gender (N = 30 total) and froze immediately after dissection and saved in −80 for further preparation.

## Microglia isolation

Rat brains were extracted from 1.5-month-old rats after intracardiac PBS perfusion. Brains were enzymatically and mechanically dissociated into a cell suspension using the Adult Brain Dissociation Kit (Miltenyi 130-107-677) and gentleMACS Octo Dissociator (Miltenyi 130-095-937). Microglia were isolated using CD11b/c magnetic microbeads (Miltenyi 130-105-634) according to the manufacturer's instructions. Microglia were snap frozen in liquid nitrogen, and later used for RNA extraction.

## RNA extraction and quantitative RT-PCR

Total brain RNA or microglia RNA was extracted with RNeasy RNA Isolation kit (Qiagen 74106) and used to generate cDNA with a High-Capacity cDNA Reverse Transcription Kit (Thermo 4368814) with random hexamer priming. For total brain RNA, frozen hemispheres were homogenized in buffer (250 mM sucrose, 20 mM Tris-base pH 7.4, 1 mM EDTA, 1 mM EGTA) and an aliquot (~40 mg) was used as input for RNA extraction. Real time polymerase chain reaction was carried out with TaqMan Fast Advanced Master Mix (Thermo 4444556), and the appropriate TaqMan (Thermo) probes. *Gapdh* was detected with probe Rn01775763_g1 (exon junctions 2-3, and 7-8) and *Treml1* was detected with Rn01511908_g1 (exon junctions 4–5). Samples were analyzed on a QuantStudio 6 Flex Real-Time PCR System (Thermo 4485697), and relative RNA amounts were quantified using Lin-RegPCR software (hartfaalcentrum.nl).

## Brain slice preparation and CSF collection

6–8 week old rats were deeply anesthetized with isoflurane, and CSF was immediately collected from the cisterna magna using customized glass pipettes connected with syringes. The rats were then intracardially perfused with an ice-cold cutting solution containing (in mM) 120 Choline Chloride, 2.6 KCl, 26 NaH CO3, 1.25 NaH2PO4, 0.5 CaCl2, 7 MgCl2, 1.3 Ascorbic Acid, 15 Glucose, pre-bubbled with 95% O2/5% CO2 for 15 min. The brains were rapidly removed from the skull. Coronal brain slices containing the hippocampal formation (400 μm thick) were prepared in the ice-cold cutting solution bubbled with 95% O2/5% CO2 using Vibratome VT1200S (Leica Microsystems, Germany) and then incubated in an interface chamber in ACSF containing (in mM): 126 NaCl, 3 KCl, 1.2 NaH2PO4; 1.3 MgCl2, 2.4 CaCl2, 26 NaHCO3, and 10 glucose (at pH 7.3), bubbled with 95% O2 and 5% CO2 at 30°C for 1 hr and then kept at room temperature. The hemi-slices were transferred to a recording chamber perfused with ACSF at a flow rate of ~2 ml/min using a peristaltic pump.

## Electrophysiological recording

Whole-cell recordings in the voltage-clamp mode ($-70$ mv) were made with patch pipettes containing (in mM): 132.5 Cs-gluconate, 17.5 CsCl, 2 MgCl2, 0.5 EGTA, 10 HEPES, 4 ATP and 5 QX-314, with pH adjusted to 7.3 by CsOH. Patch pipettes (resistance, 8–10 M$\Omega$) were pulled from 1.5 mm thin-walled borosilicate glass (Sutter Instruments, Novato, CA) on a horizontal puller (model P-97; Sutter Instruments, Novato, CA).

Basal synaptic responses were evoked at 0.05 Hz by electrical stimulation of the hippocampal SC–CA3 >CA1 afferents using concentric bipolar electrodes. CA1 neurons were viewed under upright microscopy (FN-1, Nikon Instruments, Melville, NY) and recorded with Axopatch-700B amplifier (Molecular Devices, San Jose, CA). Data were low-pass filtered at 2 kHz and acquired at 5–10 kHz. The series resistance (Rs) was consistently monitored during recording in case of reseal of ruptured membrane. Cells with Rs >20 M$\Omega$ or with Rs deviated by >20% from initial values were excluded from analysis. EPSCs were recorded in ACSF containing 15 µM bicuculline methiodide to block GABA-A receptors. The stimulation intensity was adjusted to evoke EPSCs that were 40% of the maximal evoked amplitudes ('test intensity'). 5–10 min after membrane rupture, EPSCs were recorded for 7 min at a test stimulation intensity that produced currents of ~40% maximum. For recording of paired-pulse ratio (PPR), paired-pulse stimuli with 50 ms or 200 ms inter-pulse interval were given. The PPR was calculated as the ratio of the second EPSC amplitude to the first. For recording of AMPA/NMDA ratio, the membrane potential was firstly held at-70 mV to record only AMPAR current for 20 sweeps with 20 s intervals. Then the membrane potential was turned to +40 mV to record NMDAR current for 20 sweeps with perfusion of 5 µM NBQX to block AMPAR. Mini EPSCs were recorded by maintaining neurons at $-70$ mV with ACSF containing 1 µM TTX and 15 µM bicuculline methiodide to block action potentials and GABA-A receptors respectively. mEPSCs were recorded for 5–10 mins for analysis. Data were collected and analyzed using the Axopatch 700B amplifiers and pCLAMP10 software (Molecular Devices) and mEPSCs are analyzed using mini Analysis Program.

Field potential recordings in the current-clamp mode were made with pipettes (resistance, 3–4 M$\Omega$) containing 3 M NaCl. Basal synaptic responses were evoked at 0.05 Hz by electrical stimulation of the Schaffer collateral afferents using concentric bipolar electrodes. fEPSPs were recorded from CA3-CA1 synapse area near Schaffer collateral afferents. For the input-output curve, the stimulus intensity was raised from 0.4 mA in steps of 0.2 mA. LTP was induced after 10–20 mins of baseline recording with 5 trains of 1 s 100 Hz stimulation with 5 s intervals. Following assessment of basal synaptic transmission (BST) by plotting the stimulus voltages against slopes of field Excitatory Post-Synaptic Potentials (fEPSP), baseline was recorded every minute at an intensity that evoked a response 35% of the maximum evoked response. Responses were measured as fEPSP slopes expressed as percentage of baseline.

## Antibodies treatment

10 ng/ml Goat anti-TNF-$\alpha$ (AF-510-NA, R and D Systems) or Goat IgG control (AB-108-C, R and D Systems) was incubated right after slice cutting and perfused throughout recording. Experiments were performed at $28.0 \pm 0.1°$C.

## Statistical analysis

Data were analyzed using GraphPad Prism software and expressed as mean $\pm$ s.e.m. Statistical tests used to evaluate significance are shown in Figure legends. Significant differences were accepted at $p < 0.05$.

## Additional information

### Funding

| Funder | Grant reference number | Author |
| --- | --- | --- |
| National Institute on Aging | R01AG063407 | Luciano D'Adamio |
| National Institute on Aging | RF1AG064821 | Luciano D'Adamio |

The funders had no role in study design, data collection and interpretation, or the decision to submit the work for publication.

## Author contributions
Siqiang Ren, Conceptualization, Data curation, Formal analysis, Investigation, Writing - review and editing; Wen Yao, Data curation, Formal analysis, Validation, Investigation, Methodology, Writing - review and editing; Marc D Tambini, Conceptualization, Data curation, Validation, Investigation, Writing - review and editing; Tao Yin, Investigation, Writing - review and editing; Kelly A Norris, Investigation; Luciano D'Adamio, Conceptualization, Resources, Data curation, Formal analysis, Supervision, Funding acquisition, Validation, Investigation, Visualization, Methodology, Writing - original draft, Project administration, Writing - review and editing

## Author ORCIDs
Marc D Tambini (ID) http://orcid.org/0000-0003-4461-586X
Luciano D'Adamio (ID) https://orcid.org/0000-0002-2204-9441

## Ethics
Animal experimentation: All experiments were done according to policies on the care and use of laboratory animals of the Ethical Guidelines for Treatment of Laboratory Animals of the NIH. The procedures were described and approved by the Rutgers Institutional Animal Care and Use Committee (IACUC) (protocol number 201702513). All efforts were made to minimize animal suffering and reduce the number of animals used. The animals were housed two per cage under controlled laboratory conditions with a 12 hr dark light cycle, a temperature of 22 ± 2°C. Rats had free access to standard rodent diet and tap water.

## Decision letter and Author response
Decision letter https://doi.org/10.7554/eLife.57513.sa1
Author response https://doi.org/10.7554/eLife.57513.sa2

## Additional files

### Supplementary files
• Transparent reporting form

### Data availability
All data generated or analyzed during this study are included in the Source data files have been provided for all figures.

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
