## [Decision Letter]

**Acceptance summary:**

An open question in the development of Alzheimer's disease, mainly the late onset form of the disease, is what changes occur early, prior to known pathologies that only occur in the full blown disease. Here an excellent model of the disease is used to identify changes that occur in the brain at a young age. For this, the authors have generated and used rats expressing the R47H *TREM2* mutation, along with humanized APP, to test *TREM2*-mediated cascades more appropriately.

**Decision letter after peer review:**

Thank you for submitting your article "Microglia *TREM2^R47H^* Alzheimer-linked variant enhances excitatory transmission and reduces LTP via increased TNF-α levels" for consideration by *eLife*. Your article has been reviewed by three peer reviewers, and the evaluation has been overseen by a Reviewing Editor and Satyajit Rath as the Senior Editor. The following individuals involved in review of your submission have agreed to reveal their identity: Efrat Levy (Reviewer #1); Li Gan (Reviewer #3).

The reviewers have discussed the reviews with one another and the Reviewing Editor has drafted this decision to help you prepare a revised submission.

Summary:

Due to the association of the *TREM2^47H^* mutation with increased Alzheimer's Disease (AD), *TREM2* KO and *TREM2* mutant KI mouse models have been actively investigated for the last 5 years in a wide array of neurodevelopmental, neurotrauma and neurodegeneration model systems as well as AD model systems. Here the authors generate rats expressing the *R47HTREM2* mutation. Generation of rat models is critical for testing the effects of *TREM2* mutations on a greater array of behaviors than possible in murine models and the larger size of the rat makes analysis of material such as CSF and fMRIs much more feasible with standard equipment. The authors also generated this KI on a rat model with humanized APP to more appropriately test *TREM2* mediated cascades on AB generation.

Revisions expected in follow-up-work:

*TREM2* locus is in close proximity with *Treml1*, and several other trem related genes. Previous studies have shown that *Treml1* could be affected in some *Trem2* knockout models (Kang et al., 2018). It would be helpful to measure the levels of the neighboring trem genes, especially *treml1*, using qRT-PCR.

---

## [Author Response]

Summary:Due to the association of the TREM2^47H^ mutation with increased Alzheimer's Disease (AD), TREM2 KO and TREM2 mutant KI mouse models have been actively investigated for the last 5 years in a wide array of neurodevelopmental, neurotrauma and neurodegeneration model systems as well as AD model systems. Here the authors generate rats expressing the R47H TREM2 mutation. Generation of rat models is critical for testing the effects of TREM2 mutations on a greater array of behaviors than possible in murine models and the larger size of the rat makes analysis of material such as CSF and fMRIs much more feasible with standard equipment. The authors also generated this KI on a rat model with humanized APP to more appropriately test TREM2 mediated cascades on AB generation.Revisions expected in follow-up-work:TREM2 locus is in close proximity with Treml1, and several other trem related genes. Previous studies have shown that Treml1 could be affected in some Trem2 knockout models (Kang et al., 2018). It would be helpful to measure the levels of the neighboring trem genes, especially treml1, using qRT-PCR.

Thank you for the suggestion. We have performed the suggested experiment and measured *Treml1* mRNA in both purified Microglia and total brain. The results, which show no significant differences among the three genotypes (*Trem2^w/w^,Trem2^R47H/w^* and *Trem2^R47H/R47H^*) are shown in the new Figure 4. This result is not surprising given that our genetic manipulation results in minimal alteration of the nucleotide sequence of the *Trem2* gene locus (2 bases) with no deletions. In contrast, the *Trem2* knockout mouse model used in the aforementioned paper (now referenced in the manuscript) was “generated from the Velocigene ‘definitive null’ targeting strategy in which the entire coding region was replaced by selection cassette (lacZ-flox-human Ubiquitin C promoter-neomycin-flox), beginning at 16 bp upstream of the ATG start codon and ending at the TGA stop codon of *Trem2*.”